# Homomorphic Self-Supervised Learning

**T. Anderson Keller** *t.anderson.keller@gmail.com*
*Apple*

**Xavier Suau** *xsuaucuadros@apple.com*
*Apple*

**Luca Zappella** *lzappella@apple.com*
*Apple*

**Reviewed on OpenReview:** *https://openreview.net/forum?id=tEKqQgbwbf*

## Abstract

Many state of the art self-supervised learning approaches fundamentally rely on transformations applied to the input in order to selectively extract task-relevant information. Recently, the field of equivariant deep learning has developed to introduce structure into the feature space of deep neural networks by designing them as homomorphisms with respect to input transformations. In this work, we observe that many existing self-supervised learning algorithms can be both unified and generalized when seen through the lens of equivariant representations. Specifically, we introduce a general framework we call *Homomorphic Self-Supervised Learning*, and theoretically show how it may subsume the use of input-augmentations provided an augmentation-homomorphic feature extractor. We validate this theory experimentally for simple augmentations, demonstrate the necessity of representational structure for feature-space SSL, and further empirically explore how the parameters of this framework relate to those of traditional augmentation-based self-supervised learning. We conclude with a discussion of the potential benefits afforded by this new perspective on self-supervised learning.

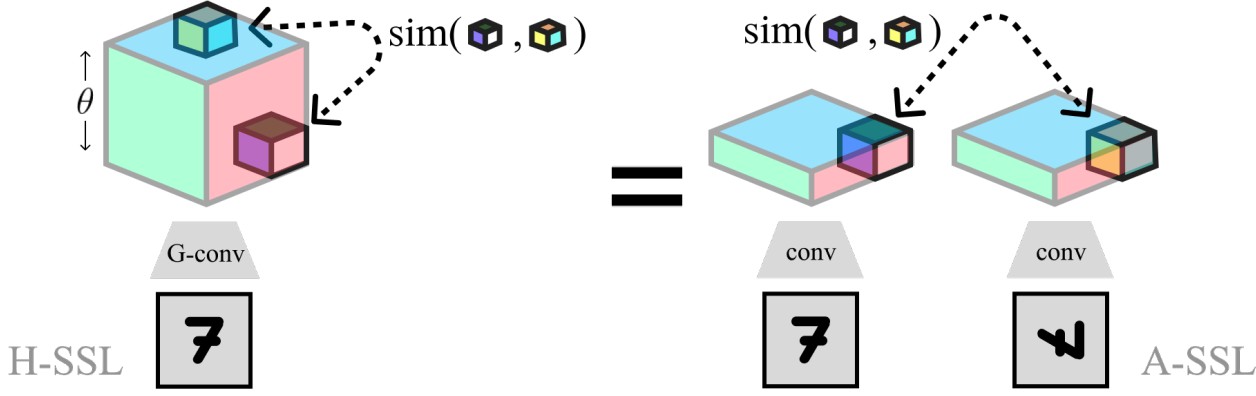

Figure 1: Overview of Homomorphic-SSL (left) and its relation to traditional Augmentation-based SSL (right). Positive pairs extracted from the lifted dimension ($\theta$) of a rotation equivariant network (G-conv) are equivalent to pairs extracted from the separate representations of two rotated images.

# 1 Introduction

Many self-supervised learning (SSL) techniques can be colloquially defined as representation learning algorithms which extract approximate supervision signals directly from the input data itself (LeCun & Misra, 2021). In practice, this supervision signal is often obtained by performing symmetry transformations of the input with respect to task-relevant information, meaning the transformations leave task-relevant information unchanged, while altering task-irrelevant information. Numerous theoretical and empirical works have shown that by combining such symmetry transformations with contrastive objectives, powerful lower dimensional representations can be learned which support linear-separability (Wang et al., 2022; Lee et al., 2021; Tian et al., 2020b; Arora et al., 2019; Tosh et al., 2020), identifiability of generative factors (von Kügelgen et al., 2021; Tsai et al., 2020; Federici et al., 2020; Ji et al., 2021), and reduced sample complexity (Grill et al., 2020; Chen et al., 2020).

One rapidly developing domain of deep learning research which is specifically focused on learning structured representations of the input with respect to symmetry transformations is that of equivariant neural networks (Cohen & Welling, 2016; 2017; Weiler et al., 2018; Worrall & Welling, 2019; Finzi et al., 2020; 2021; van der Pol et al., 2020). Formally, equivariant neural networks are designed to be group homomorphisms for transformation groups which act on the input space – meaning that their output explicitly preserves the structure of the input with respect to these transformations. Traditionally, equivariant neural networks have been contrasted with data augmentation in the supervised setting, and proposed as a more robust and data efficient method for incorporating prior symmetry information into deep neural networks (Worrall et al., 2017). In the self-supervised setting, however, where data augmentation is itself implicitly responsible for extracting the supervision signal from the data, the relationship between data augmentation and equivariance is much more nuanced.

In this work, we study self-supervised learning algorithms when equivariant neural networks are used as backbone feature extractors. Interestingly, we find a convergence of existing loss functions from the literature, and ultimately generalize these with the framework of *Homomorphic Self-Supervised Learning*. Experimentally, we show that, when the assumption of an augmentation-homomorphic backbone is satisfied, this framework subsumes input augmentation, as evidenced by identical performance over a range of settings. We further validate this theory by showing that when our assumption is not satisfied, the framework fails to learn useful representations. Finally, we explore the new generalized parameters introduced by this framework, demonstrating an immediate path forward for improvements to existing SSL methods which operate without input augmentations.

# 2 Background

In this section, we introduce the concept of equivariance and show how structured representations can be obtained via $\mathcal{G}$-convolutions (Cohen & Welling, 2016). We then review general self-supervised frameworks and how prior literature differs with respect to its use of input augmentations.

## 2.1 Equivariance

Formally, a map which preserves the structure of the input space in the output space is termed a homomorphism. The most prominent example of a homomorphism in modern deep learning is the class of group equivariant neural networks, which are analytically constrained to be group homomorphisms for specified transformation groups (such as translation, rotation, mirroring, and scaling). The map $f : \mathcal{X} \to \mathcal{Z}$ is said to be equivariant with respect to the group $\mathcal{G} = (G, \cdot)$ if

$$\exists \Gamma_g \quad \text{such that} \quad f(T_g[\boldsymbol{x}]) = \Gamma_g[f(\boldsymbol{x})] \quad \forall g \in G , \tag{1}$$

where $G$ is the set of all group elements, $\cdot$ is the group operation, $T_g$ is the representation of the transformation $g \in G$ in input space $\mathcal{X}$, and $\Gamma_g$ is the representation of the same transformation in output space $\mathcal{Z}$. If $T_g$ and $\Gamma_g$ are formal group representations (Serre, 1977) such maps $f$ are termed group-homomorphisms since they preserve the structure of the group representation $T_g$ in input space with the output representation $\Gamma_g$. There are many different methods for constructing group equivariant neural networks, resulting in different

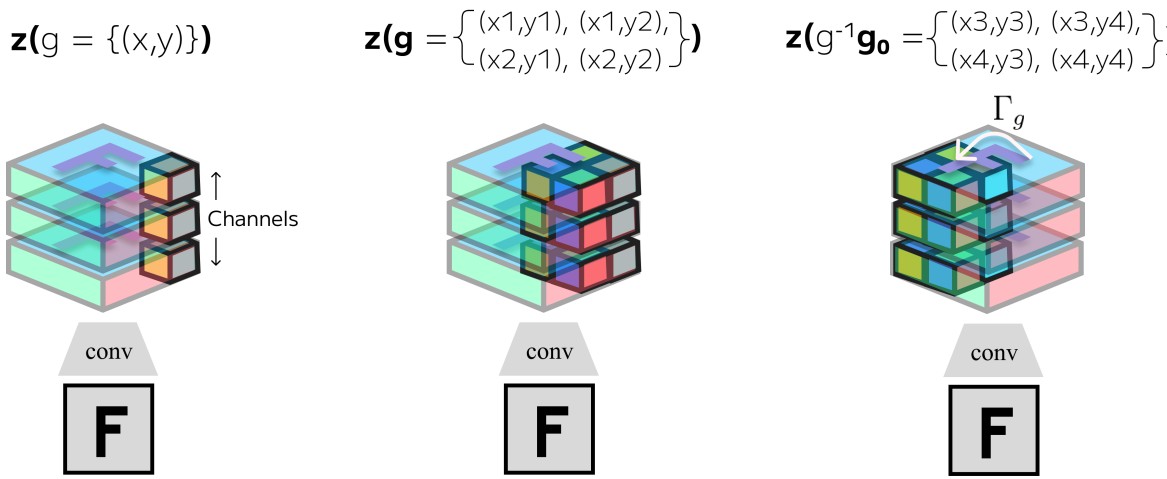

Figure 2: Visualization of a 'fiber' (left), a 'fiber bundle' (center) and a group representation $\Gamma_g$ acting on a fiber bundle (right). We see that a fiber is defined as all features at an individual group element (in this case all feature channels at an individual spatial dimension), while a fiber bundle is all features at a set of ordered group elements. In this figure, we depict feature channels stacked along the z-dimension, different from the 'lifted' dimension in Figure 1 (left).

representations of the transformation in feature space $\Gamma_g$. In this work, we consider only discrete groups $\mathcal{G}$ and networks which admit regular representations for $\Gamma$. In the following paragraph we outline one method by which such networks can be built, and thereby demonstrate how the regular representation behaves.

**Group-Convolutional Neural Networks** One common way in which group-equivariant networks are constructed is via the group-convolution (G-conv) (Cohen & Welling, 2016). For a discrete group $\mathcal{G}$, we denote the pre-activation output of a $\mathcal{G}$-equivariant convolutional layer $l$ as $\boldsymbol{z}^l$, with a corresponding input $\boldsymbol{y}^l$. In practice these values are stored in finite arrays with a feature multiplicity equal to the order of the group in each space. Explicitly, $\boldsymbol{z}^l \in \mathbb{R}^{C_{out} \times |G_{out}|}$, and $\boldsymbol{y}^l \in \mathbb{R}^{C_{in} \times |G_{in}|}$ where $G_{out}$ and $G_{in}$ are the set of group elements in the output and input spaces respectively. We use the following shorthand for indexing $\boldsymbol{z}^l(g) \equiv \boldsymbol{z}^{l,:,g} \in \mathbb{R}^{C_{out}}$ and $\boldsymbol{y}^l(g) \equiv \boldsymbol{y}^{l,:,g} \in \mathbb{R}^{C_{in}}$, denoting the vector of feature channels at a specific group element (sometimes called a 'fiber' (Cohen & Welling, 2017)). Then, the value $z^{l,c}(g) \in \mathbb{R}$ of a single output at layer $l$, channel $c$ and element $g$ is

$$z^{l,c}(g) \equiv [\boldsymbol{y}^l \star \boldsymbol{\psi}^{l,c}](g) = \sum_{h \in G_{in}} \sum_i^{C_{in}} y^{l,i}(h) \psi_i^{l,c}(g^{-1} \cdot h) \;, \tag{2}$$

where $\boldsymbol{\psi}_i^{l,c}$ is the filter between the $i^{th}$ input channel (subscript) and the $c^{th}$ output channel (superscript), and is similarly defined (and indexed) over the set of input group elements $G_{in}$. We highlight that the composition $g^{-1} \cdot h = k \in G_{in}$ is defined by the action of the group and yields another group element by closure of the group. The representation $\Gamma_g$ and can then be defined as $\Gamma_g[\boldsymbol{z}^l(h)] = \boldsymbol{z}^l(g^{-1} \cdot h)$ for all $l > 0$ when $\mathcal{G}_{in}^l = \mathcal{G}_{out}^l = \mathcal{G}_{out}^0$. From this definition it is straightforward to prove equivariance from: $[\Gamma_g[\boldsymbol{y}^l] \star \boldsymbol{\psi}^l](h) = \Gamma_g[\boldsymbol{y}^l \star \boldsymbol{\psi}^l](h) = \Gamma_g[\boldsymbol{z}^l](h)$. Furthermore, we see that $\Gamma_g$ is a 'regular representation' of the group, meaning that it acts by permuting features along the group dimension while leaving feature channels intact. Group equivariant layers can then be composed with pointwise non-linearities and biases to yield a fully equivariant deep neural network (e.g. $\boldsymbol{y}_i^{l+1} = \text{ReLU}(\boldsymbol{z}^l + \boldsymbol{b})$ where $\boldsymbol{b} \in \mathbb{R}^{C_{out}}$ is a learned bias shared over the output group dimensions). For $l = 0$, $\boldsymbol{y}^0$ is set to the raw input $\boldsymbol{x}$, and typically the input group is set to the group of all 2D integer translations up to height $H$ and width $W$: $\mathcal{G}_{in}^0 = (\mathbb{Z}_{HW}^2, +)$. The output group $\mathcal{G}_{out}^0$ is then chosen by the practitioner and is typically a larger group which includes translation as a subgroup, e.g. the roto-translation group, or the group of scaling & translations. In this way, the first layer of a group-equivariant neural network is frequently called the 'lifting layer' since it lifts the input from

the translation group, containing only spatial dimensions, to a larger group by adding an additional 'lifted' dimension.

**Example**  As a simple example, a standard convolutional layer would have all height ($H$) and width ($W$) spatial coordinates as the set $G_{out}$, giving $\boldsymbol{z} \in \mathbb{R}^{C \times HW}$. A group-equivariant neural network (Cohen & Welling, 2016) which is equivariant with respect to the the group of all integer translations and 90-degree rotations ($p4$) would thus have a feature multiplicity four times larger ($\boldsymbol{z} \in \mathbb{R}^{C \times 4HW}$), since each spatial element is associated with the four distinct rotation elements ($0^o, 90^o, 180^o, 270^o$). Such a rotation equivariant network is depicted in Figure 1 with the 'lifted' rotation dimension extended along the vertical axis ($\theta$). In both the translation and rotation cases, the regular representation $\Gamma_g$ acts by permuting the representation along the group dimension, leaving the feature channels unchanged.

**Notation**  In the remainder of this paper we will see that it is helpful to have a notation which allows for easy reference to the sets of features corresponding to multiple group elements simultaneously. These sets are sometimes called 'fiber bundles' and are visually compared with individual fibers in Figure 2. In words, a fiber (left) can be described as all features values at a specific group element (such as all channels at a given spatial location), and a fiber bundle (center) is then all features at an ordered set of group elements (such as all channels for a given spatial patch). We denote the set of fibers corresponding to an ordered set of group elements $\boldsymbol{g}$ as: $\boldsymbol{z}(\boldsymbol{g}) = [\boldsymbol{z}(g) \mid g \in \boldsymbol{g}] \in \mathbb{R}^{|\boldsymbol{g}|C_{out}}$. Using this notation, we can define the action of $\Gamma_g$ as: $\Gamma_g[\boldsymbol{z}(\boldsymbol{g}_0)] = \boldsymbol{z}(g^{-1} \cdot \boldsymbol{g}_0)$. Thus $\Gamma_g$ can be seen to move the fibers from 'base' locations $\boldsymbol{g}_0$ to a new ordered set of locations $g^{-1} \cdot \boldsymbol{g}_0$, as depicted in on the right side of Figure 2. We highlight that order is critical for our definition since a transformation such as rotation may simply permute $\boldsymbol{g}_0$ while leaving the unordered set intact.

## 2.2   Self-Supervised Learning

As mentioned in Section 1, self-supervised learning can be seen as extracting a supervision signal from the data itself, often by means of transformations applied to the input. Many terms in self-supervised learning objectives can thus often be abstractly written as a function $\mathcal{I}(\boldsymbol{V}^{(1)}, \boldsymbol{V}^{(2)})$ of two batches of vectors $\boldsymbol{V}^{(1)} = \{\boldsymbol{v}_i^{(1)}\}_{i=1}^N$ and $\boldsymbol{V}^{(2)} = \{\boldsymbol{v}_i^{(2)}\}_{i=1}^N$ where there is some relevant relation between the elements of the two batches. In this description, we see that there are two main degrees of freedom which we will explore in the following paragraphs: the choice of function $\mathcal{I}$, and the precise relationship between $\boldsymbol{V}^{(1)}$ and $\boldsymbol{V}^{(2)}$.

**SSL Loss Functions: $\mathcal{I}^{\mathbf{C}}$ and $\mathcal{I}^{\mathbf{NC}}$**  The most prominent SSL loss terms in the literature are often segregated into contrastive $\mathcal{I}^{\mathrm{C}}$ (Chen et al., 2020; Oord et al., 2018) and non-contrastive $\mathcal{I}^{\mathrm{NC}}$ (Grill et al., 2020; Chen & He, 2020) losses. At a high level, *contrastive losses* frequently rely on a vector similarity function $\mathrm{sim}(\cdot, \cdot)$ (such as cosine similarity), and 'contrast' its output for 'positive' and 'negative' pairs. A general form of a contrastive loss, inspired by the 'InfoNCE' loss (Oord et al., 2018), can be written as:

$$\mathcal{I}_i^{\mathrm{C}}(\boldsymbol{V}^{(1)}, \boldsymbol{V}^{(2)}) = -\frac{1}{N} \log \frac{\exp\Big(\mathrm{sim}\big(h(\boldsymbol{v}_i^{(1)}), h(\boldsymbol{v}_i^{(2)})\big)/\tau\Big)}{\sum_{j \neq i}^N \sum_{k,l}^2 \exp\Big(\mathrm{sim}\big(h(\boldsymbol{v}_i^{(k)}), h(\boldsymbol{v}_j^{(l)})\big)/\tau\Big)} \tag{3}$$

where $h$ is a non-linear 'projection head' $h : \mathcal{Z} \to \mathcal{Y}$ and $\tau$ is the 'temperature' of the softmax. We see that such losses can intuitively be thought of as trying to classify the correct 'positive' pair (given by $\boldsymbol{v}_i^{(1)}$ & $\boldsymbol{v}_i^{(2)}$) out of a set of negative pairs (given by all other pairs in the batch). Comparatively, *non-contrastive losses* are often applied to the same sets of representations $\boldsymbol{V}^{(1)}$ and $\boldsymbol{V}^{(2)}$, but crucially forego the need for 'negative pairs' through other means of regularization (such as a stop-gradient on one branch (Chen & He, 2020; Tian et al., 2021) observed to regularize the eigenvalues of the representation covariance matrix). Fundamentally this often results in a loss of the form:

$$\mathcal{I}_i^{\mathrm{NC}}(\boldsymbol{V}^{(1)}, \boldsymbol{V}^{(2)}) = -\frac{1}{N} \mathrm{sim}\big(h(\boldsymbol{v}_i^{(1)}), \mathrm{SG}(\boldsymbol{v}_i^{(2)})\big) , \tag{4}$$

where SG denotes the stop-gradient operation. In this work we focus the majority of our experiments on the $\mathcal{I}^{\mathrm{NCE}}$ loss specifically. However, given this general formulation which decouples the specific loss from the

choice of pairs $\boldsymbol{V}^{(1)}$ & $\boldsymbol{V}^{(2)}$, and the fact that our framework only operates on the formulation of the pairs, we will see that our analyses and conclusions extend to all methods which can be written this way. In the following, we will introduce the second degree of freedom which captures many SSL algorithms: the precise relationship between $\boldsymbol{V}^{(1)}$ and $\boldsymbol{V}^{(2)}$.

**Relationship Between SSL Pairs: $\boldsymbol{V}^{(1)}$ & $\boldsymbol{V}^{(2)}$**  Similar to our treatment of SSL loss functions $\mathcal{I}$, in this section we separate the existing literature into two categories with respect to the leveraged relationship between positives pairs. Specifically, we compare methods which rely on input augmentations, which we call Augmentation-based SSL (A-SSL), to methods which operate entirely within the representation of a single input, which we call Feature-space SSL (F-SSL). An influential framework which relies on augmentation is the SimCLR framework (Chen et al., 2020). Using the above notation, this is given as:

$$\mathcal{L}_i^{\text{A-SSL}}(\mathbf{X}) = \underset{g_1, g_2 \sim G}{\mathbb{E}} \mathcal{I}_i^{\text{C}}\left(\left\{f\left(T_{g_1}[\boldsymbol{x}_n]\right)\right\}_n^N, \left\{f\left(T_{g_2}[\boldsymbol{x}_n]\right)\right\}_n^N\right), \tag{5}$$

where $T_g[\boldsymbol{x}]$ denotes the action of the sampled augmentation $g$ on the input, $G$ is the set of all augmentations, and $f(\boldsymbol{x}) = \boldsymbol{v}$ is the backbone feature extractor to be trained. This loss is then summed over all elements $i$ in the batch before backpropagation. In this work, we consider this SimCLR loss given in Equation 5 as the canonical A-SSL method given its broad adoption and similarity with other augmentation-based methods. The second class of SSL methods we consider in this work are those which operate without the use of explicit input augmentations, but instead compare subsets of a representation for a single image directly. Models such as Deep InfoMax (DIM(L)) (Hjelm et al., 2019), Greedy InfoMax (GIM) (Löwe et al., 2019), and Contrastive Predictive Coding (CPC) (Oord et al., 2018)[1] can all be seen to be instantiations of such Feature-space SSL methods. At a low level, these methods vary in the specific subsets of the representations which are used in the loss (from single spatial elements to larger 'patches'), and vary in the similarity function (with some using a log-bilinear model $\text{sim}(\boldsymbol{a}, \boldsymbol{b}) = \exp\left(\boldsymbol{a}^T W \boldsymbol{b}\right)$, instead of cosine similarity). In this work we define a general Feature-space SSL (F-SSL) loss in the spirit of these models which similarly operates in the feature space of a single image, uses an arbitrary spatial 'patch' size $|\boldsymbol{g}|$, and a cosine similarity function. Formally:

$$\mathcal{L}_i^{\text{F-SSL}}(\mathbf{X}) = \underset{\boldsymbol{g}_1, \boldsymbol{g}_2 \sim \mathbb{Z}_{HW}^2}{\mathbb{E}} \mathcal{I}_i^{\text{C}}\left(\left\{\boldsymbol{z}_n\left(\boldsymbol{g}_1\right)\right\}_n^N, \left\{\boldsymbol{z}_n\left(\boldsymbol{g}_2\right)\right\}_n^N\right), \tag{6}$$

where $\boldsymbol{g} \sim \mathbb{Z}_{HW}^2$ refers to sampling a contiguous patch from the spatial coordinates of a convolutional feature map, and $\boldsymbol{z}_n$ is the output of our backbone $f(\boldsymbol{x}_n)$. In the following section, we show how equivariant backbones unify these two losses into a single loss, helping to explain both their successes and limitations while additionally demonstrating clear directions for their generalization.

## 3 Homomorphic Self-Supervised Learning

In this section we introduce Homomorphic Self-Supervised Learning (H-SSL) as a general framework for SSL with homomorphic encoders, and further show it both generalizes and unifies many existing SSL algorithms.

To begin, consider an A-SSL objective such as Equation 5 when $f$ is equivariant with respect to the input augmentation. By the definition of equivariant maps in Equation 1, the augmentation commutes with the feature extractor: $f(T_g[\boldsymbol{x}]) = \Gamma_g[f(\boldsymbol{x})]$. Thus, replacing $f(\boldsymbol{x}_n)$ with its output $\boldsymbol{z}_n = \boldsymbol{z}_n(\boldsymbol{g}_0)$, and applying the definition of the operator, we get:

$$\mathcal{L}_i^{\text{H-SSL}}(\mathbf{X}) = \underset{g_1, g_2 \sim G}{\mathbb{E}} \mathcal{I}_i^{\text{C}}\left(\left\{\boldsymbol{z}_n\left(g_1^{-1} \cdot \boldsymbol{g}_0\right)\right\}_n^N, \left\{\boldsymbol{z}_n\left(g_2^{-1} \cdot \boldsymbol{g}_0\right)\right\}_n^N\right). \tag{7}$$

Ultimately, we see that $\mathcal{L}^{\text{H-SSL}}$ subsumes the use of input augmentations by defining the 'positive pairs' as two fiber bundles from *the same representation $\boldsymbol{z}_n$*, simply indexed using two differently transformed

---

[1]In CPC, the authors use an autoregressive encoder to encode one element of the positive pairs. In GIM, they find that in the visual domain, this autoregressive encoder is not necessary, and thus the loss reduces to simple contrasting the representations from raw patches with one another, as defined here.

base spaces $g_1^{-1} \cdot \boldsymbol{g}_0$ and $g_2^{-1} \cdot \boldsymbol{g}_0$ (depicted in Figure 1, and Figure 2, center & right). Interestingly, this loss highlights the base space $\boldsymbol{g}_0$ as a parameter choice previously unexplored in the A-SSL frameworks. In Section 4 we empirically explore different choices of $\boldsymbol{g}_0$ and comment on their consequences.

A second interesting consequence of this derivation is the striking similarity of the $\mathcal{L}^{\text{H-SSL}}$ objective and other existing SSL objectives which operate without explicit input augmentations to generate multiple views. This can be seen most simply by comparing $\mathcal{L}^{\text{H-SSL}}$ from Equation 7 with the $\mathcal{L}^{\text{F-SSL}}$ objective from Equation 6. Specifically, since $\boldsymbol{g}_1$ & $\boldsymbol{g}_2$ from the F-SSL loss can be decomposed as a single base patch $\boldsymbol{g}_0$ offset by two single translation elements $g_1$ & $g_2$ (e.g. $\boldsymbol{g}_1 = g_1^{-1}\boldsymbol{g}_0$ and $\boldsymbol{g}_2 = g_2^{-1}\boldsymbol{g}_0$), we see that Equation 6 can be derived directly from Equation 7 by setting $G = \mathbb{Z}_{HW}^2$ and the size of the base patch $|\boldsymbol{g}_0|$ equal to the size of the patches used for each F-SSL case. Consequently, these F-SSL losses are contained in our framework where the set of 'augmentations' ($\mathcal{G}$) is the 2D translation group, and the base space ($\boldsymbol{g}_0$) is a small subset of the spatial coordinates. Since $\mathcal{L}^{\text{H-SSL}}$ is also derived directly from $\mathcal{L}^{\text{A-SSL}}$ (when $f$ is equivariant), we see that it provides a means to unify these previously distinct sets of SSL objectives. In Section 4 we validate this theoretical equivalence empirically. Furthermore, since $\mathcal{L}^{\text{H-SSL}}$ is defined for transformation groups beyond translation, it can be seen to generalize F-SSL objectives in a way that we have not previously seen exploited in the literature. In Section 4 we include a preliminary exploration of this generalization to scale and rotation groups.

## 4 Experiments

In this section, we empirically validate the derived equivalence of A-SSL and H-SSL in practice, and further reinforce our stated assumptions by demonstrating how H-SSL objectives (and by extension F-SSL objectives) are ineffective when representational structure is removed. We study how the parameters of H-SSL (topographic distance) relate to those traditionally used in A-SSL (augmentation strength), and finally explore how the new parameter generalizations afforded by our framework (such as choices of $\boldsymbol{g}_0$ and $\mathcal{G}$) impact performance.

### 4.1 Empirical Validation

For perfectly equivariant networks $f$, and sets of transformations which exactly satisfy the group axioms, the equivalence between Equations 5 and 7 is exact. However, in practice, due to aliasing, boundary effects, and sampling artifacts, even for simple transformations such as translation, equivariance has been shown to not be strictly satisfied (Zhang, 2019). In Table 1 we empirically validate our proposed theoretical equivalence between $\mathcal{L}^{\text{A-SSL}}$ and $\mathcal{L}^{\text{H-SSL}}$, showing a tight correspondence between the downstream accuracy of linear classifiers trained on representations learned via the two frameworks.

Precisely, for each transformation (Rotation, Translation, Scale), we use a backbone network which is equivariant specifically with respect to that transformation (e.g. rotation equivariant CNNs, regular CNNs, and Scale Equivariant Steerable Networks (SESN) (Sosnovik et al., 2020)). For A-SSL we augment the input at the pixel level by: randomly translating the image by up to $\pm\,20\%$ of its height/width (for translation), randomly rotating the image by one of $[0^o, 90^o, 180^o, 270^o]$ (for rotation), or randomly downscaling the image to a value between 0.57 & 1.0 of its original scale. These two augmented versions of the image are then fed through the backbone separately, and a single fiber (meaning $|\boldsymbol{g}_0| = 1$) is randomly selected. Although the fiber is selected from the same base location for both images, it will contain different features since the underlying images are transformed differently. We investigate the impact of the base space size separately in Section 4.3. For H-SSL we use no input augmentations and instead rely on differently indexed base patches (formed by shifting the randomly selected fiber $\boldsymbol{g}_0$ by two separate randomly selected group elements $g_1$ & $g_2$). For example, for A-SSL with translation, we compare the feature vectors for two translated images *at the same pixel location* $\boldsymbol{g}_0$. For H-SSL with translation, we compare the feature vectors of a single image at two *translated locations* $g_1^{-1}\cdot\boldsymbol{g}_0$ & $g_2^{-1}\cdot\boldsymbol{g}_0$. Importantly, we note that these feature vectors are taken before performing any pooling over the group dimensions in all cases. Ultimately, we see an equivalence between the performance of the A-SSL models and H-SSL models which significantly differs from the frozen and supervised baselines, validating our theoretical conclusions from Section 3. Further details on this experimental setup can be found in Appendix A.

Table 1: MNIST (LeCun & Cortes, 2010), CIFAR10 (Krizhevsky et al.) and Tiny ImageNet (Le & Yang, 2015) top-1 test accuracy (mean ± std. over 3 runs) of a detached classifier trained on the representations from SSL methods with different backbones. We compare A-SSL and H-SSL with random frozen and fully supervised backbones. We see equivalence between A-SSL and H-SSL from the first two columns, as desired, and often see a significant improvement in performance for H-SSL methods when moving from Translation to generalized groups such as Scale.

| Dataset | Transformation | Backbone | A-SSL | H-SSL | Frozen | Supervised |
|---|---|---|---|---|---|---|
| MNIST | Rotation | Rot-Eq. | $68.2 \pm 2.5$ | $70.3 \pm 5.4$ | *87.2 ± 0.8* | *99.4 ± 0.1* |
| | Translation | CNN | $95.9 \pm 0.3$ | $96.0 \pm 1.3$ | *94.1 ± 0.3* | *99.2 ± 0.1* |
| | Scale | SESN | $98.6 \pm 0.1$ | $98.3 \pm 0.2$ | *94.7 ± 0.6* | *99.3 ± 0.1* |
| CIFAR10 | Rotation | Rot-Eq. | $46.1 \pm 0.6$ | $48.3 \pm 0.5$ | *38.4 ± 0.1* | *73.0 ± 1.1* |
| | Translation | CNN | $39.2 \pm 0.5$ | $36.3 \pm 1.1$ | *40.4 ± 0.2* | *76.2 ± 1.4* |
| | Scale | SESN | $59.4 \pm 0.2$ | $56.7 \pm 0.4$ | *41.1 ± 0.6* | *78.0 ± 0.2* |
| Tiny ImageNet | Rotation | Rot-Eq. | $14.9 \pm 0.3$ | $13.5 \pm 0.5$ | *6.1 ± 0.2* | *22.5 ± 0.1* |
| | Scale | SESN | $16.2 \pm 0.4$ | $14.0 \pm 1.3$ | *6.4 ± 0.2* | *23.7 ± 0.2* |

Table 2: An extension of Table 1 with non-equivariant backbones. We see that the H-SSL methods perform similar to, or worse than, the frozen baseline when equivariance is removed, as expected.

| Dataset | Transformation | Backbone | A-SSL | H-SSL | Frozen | Supervised |
|---|---|---|---|---|---|---|
| MNIST | Translation | MLP | $87.6 \pm 0.2$ | $58.2 \pm 0.5$ | $83.0 \pm 0.8$ | $98.6 \pm 0.1$ |
| | Scale | CNN ($6 \times CHW$) | $95.2 \pm 0.1$ | $87.2 \pm 2.4$ | $87.2 \pm 0.6$ | $99.3 \pm 0.1$ |
| CIFAR10 | Scale | CNN ($6 \times CHW$) | $53.6 \pm 0.2$ | $37.5 \pm 0.1$ | $43.6 \pm 0.3$ | $67.9 \pm 2.1$ |

## 4.2 H-SSL Without Structure

To further validate our assertion that $\mathcal{L}^{\text{H-SSL}}$ requires a homomorphism, in Table 2 we show the same models from Table 1 without equivariant backbones. Explicitly, we use the same overall model architectures but replace the individual layers with non-equivariant counterparts. Specifically, for the MLP, we replace the convolutional layers with fully connected layers (slightly reducing the total number of activations from 6272 to 2048 to reduce memory consumption), and replace the SESN kernels of the scale-equivariant models with fully-parameterized, non-equivariant counterparts, otherwise keeping the output dimensionality the same (resulting in the $6 \times$ larger output dimension). Furthermore, for these un-structured representations, in the H-SSL setting, we 'emulate' a group dimension to sample 'fibers' from. For the MLP we do this by reshaping the 2048 dimensional output to (16, 128), and select one of the 16 rows at each iterations. For the CNN, we similarly use the 6 times larger feature space to sample $\frac{1}{6}^{th}$ of the elements as if they were scale-equivariant.

We thus observe that when equivariance is removed, but all else remains equal, $\mathcal{L}^{\text{H-SSL}}$ models perform significantly below their input-augmentation counterparts, and similarly to a 'frozen' randomly initialized backbone baselines, indicating the learning algorithm is no longer effective. Importantly, this indicates why existing F-SSL losses (such as DIM(L) (Hjelm et al., 2019)) always act within equivariant dimensions (e.g. between the spatial dimensions of feature map pixels) – these losses are simply ineffective otherwise. Interestingly, this provides novel insights into how the successes of early self-supervised methods may have been crucially dependent on equivariant backbones, potentially unbeknownst to the authors at the time. An intuitive understanding of this result can be given by viewing arbitrary features as being related by some unknown input transformation which may not preserve the target information about the input. In contrast, however, since equivariant dimensions rely on symmetry transforms, contrast over such dimensions is known to be equivalent to contrasting transformed inputs.

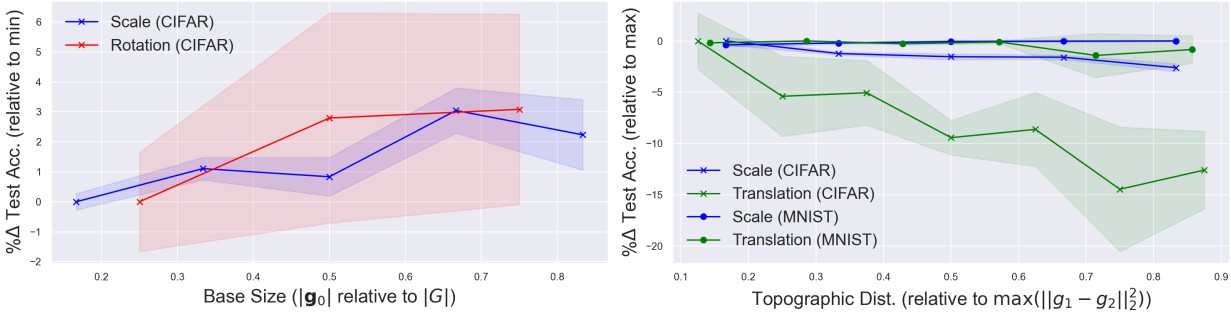

Figure 3: Study of the impact of new H-SSL parameters on top-1 test accuracy. (Left) Test accuracy marginally increases as we increase total base space size $\boldsymbol{g}_0$. (Right) Test accuracy is constant or decreases as we increase the maximum distance between fiber bundles considered positive pairs.

## 4.3 Parameters of H-SSL

**Base size** $|\boldsymbol{g}_0|$   As discussed in Section 3, The H-SSL framework identifies new parameter choices such as the base space $\boldsymbol{g}_0$. This parameter specifically carries special importance since it is the main distinction between the A-SSL and F-SSL losses in the literature. Specifically, the size of $\boldsymbol{g}_0$ is set to the full representation size in the SimCLR framework, while it is typically a small patch or an individual pixel in F-SSL losses such as DIM(L) or GIM. To investigate the impact of this difference, we explore the performance of the H-SSL framework as we gradually increase the size of $\boldsymbol{g}_0$ from 1 (akin to DIM(L) losses) to $|G| - 1$ (akin to SimCLR), with no padding. In each setting, we similarly increase the dimensionality of the input layer for the non-linear projection head $h$ to match the multiplicative increase in the dimension of the input representation $\boldsymbol{z}(\boldsymbol{g})$. In Figure 3 (left) we plot the %-change in top-1 accuracy on CIFAR-10 for each size. We see a minor increase in performance as we increase the size, but note relative stability, again suggesting greater unity between A-SSL and H-SSL.

**Topographic Distance**   Each augmentation in a standard SimCLR augmentation stack is typically associated with a scalar or vector valued 'strength'. For example, this can correspond to the maximum number of pixels translated, the range of rescaling, or the maximum number of degrees to be rotated. We note that the same concept is present in the H-SSL framework and is defined by the associated latent representation of the transformation. For networks which use regular representations (as in this work), the degree of a transformation corresponds exactly to the degree of shift within the representation. We thus propose that an analagous concept to augmentation strength is *topographic distance* in equivariant networks, meaning the distance between the two sampled fiber bundles as computed along the group dimensions (i.e. the 'degree of shift'). For example, for convolution, this would correspond to the number of feature map pixels between two patches. For scale, this would correspond to the number of scales between two patches. In Figure 3 (right), we explore how the traditional notion of augmentation 'strength' can be equated with the 'topographic distance' between $g_1$ and $g_2$ and their associated fibers (with a fixed base size of $|\boldsymbol{g}_0| = 1$). Here we approximate topographic distance as the maximum euclidean distance between sampled group elements for simplicity ($||g_1 - g_2||_2^2$), where a more correct measure would be computed using the topology of the group. We see, in alignment with prior work (Tian et al., 2020a; 2019), that the strength of augmentation (and specifically translation distance) is an important parameter for effective self supervised learning, likely relating to the mutual information between fibers as a function of distance. We believe that the reason that we do not see the inverted U-shaped relationship between accuracy and topographic distance as found by Tian et al. (2020a) is that their models have been trained on the much higher resolution DIV2K dataset, allowing for patch offsets of up to 384 pixels. In our case, since we are working on the latent representations of low resolution MNIST and CIFAR10 images, we only have a maximum offset of 8-pixels, and therefore believe that we are only capturing a fraction of the curve illustrated by others.

### 4.4 Methods

**Model Architectures** All models presented in this paper are built using the convolutional layers from the SESN (Sosnovik et al., 2020) library for consistency and comparability. For scale equivariant models, we used the set of 6 scales $[1.0, 1.25, 1.33, 1.5, 1.66, 1.75]$. To construct the rotation equivariant backbones, we use only a single scale of $[1.0]$ and augment the basis set with four 90-degree rotated copies of the basis functions at $[0^o, 90^o, 180^o, 270^o]$. These rotated copies thus defined the group dimension. This technique of basis or filter-augmentation for implementing equivariance is known from prior work and has been shown to be equivalent to other methods of constructing group-equivariant neural networks (Li et al., 2021). For translation models, we perform no basis-augmentation, and again define the set of scales used in the basis to be a single scale $[1.0]$, thereby leaving only the spatial coordinates of the final feature maps to define the output group. On MNIST (LeCun & Cortes, 2010), we used a backbone network $f$ composed of three SESN convolutional layers with 128 final output channels, ReLU activations and BatchNorms between layers. The output of the final ReLU is then considered our $z$ for contrastive learning (for $\mathcal{L}^{\text{A-SSL}}$ and $\mathcal{L}^{\text{H-SSL}}$) and is of shape $(128, S \times R, 8, 8)$ where $S$ is the number of scales for the experiment (either 1 or 6), and $R$ is the number of rotation angles (either 1 or 4). On CIFAR10 and Tiny ImageNet we used SESN-modified ResNet18 and ResNet20 models respectively where the output of the last ResNet blocks were taken as $z$ for contrastive learning. For all models where translation is not the studied transformation, we average pool over the spatial dimensions to preserve consistent input-dimensionality to the nonlinear projection head.

**Training Details** For training, we use the LARS optimizer (You et al., 2017) with an initial learning rate of 0.1, and a batch size of 4096 for all models. We use an NCE temperature ($\tau$) of 0.1, half-precision training, a learning rate warm-up of 10 epochs, a cosine lr-update schedule, and weight decay of $1 \times 10^{-4}$. On MNIST we train for 500 epochs and on CIFAR10 and Tiny ImagNet (TIN) we train for 1300 epochs.

**Computational Complexity** On average each MNIST run took 1 hour to complete distributed across 8 GPUs, and each CIFAR10/TIN run took 10 hours to complete distributed across 64 GPUs. In total this amounted to roughly 85,000 GPU hours. While equivariant models can be slightly more computationally expensive due to expanded feature space dimensionality, this cost is typically not prohibitive for training on full scale datasets such as ImageNet. In practice, we found that equivariant models did not train more than a factor of 2-3 slower than their non-equivariant counterparts, dependent on architecture, and often nearly equivalently as fast as non-equivariant baselines. Conversely, however, H-SSL may provide computational complexity reductions in certain settings, since there is not a need to perform two forward passes on augmented images, and rather all contrastive learning can be performed in the feature space of a single image. With this new framework, sufficient future work on layerwise contrastive losses, or new contrastive losses, may uncover computationally cheaper SSL algorithms leveraging H-SSL.

## 5 Related Work

Our work is built upon the literature from the fields equivariant deep learning and self-supervised learning as outlined in Sections 1 and 2. Beyond this background, our work is highly related in motivation to a number of studies specifically related to equivariance in self-supervised learning.

**Undesired Invariance in SSL** One subset of recent prior work has focused on the undesired invariances learned by A-SSL methods (Xiao et al., 2021; Tsai et al., 2022) and on developing methods by which to avoid this through learned approximate equivariance (Dangovski et al., 2022; Wang et al., 2021). Our work is, to the best of our knowledge, the first to suggest and validate that the primary reason for the success of feature-space SSL objectives such as DIM(L) (Hjelm et al., 2019) and GIM (Löwe et al., 2019) is due to their exploitation of (translation) equivariant backbones (i.e. CNNs). Furthermore, while prior work shows benefits to existing augmentation-based SSL objectives when equivariance is induced, our work investigates how equivariant representations can directly be leveraged to formulate new theoretically-grounded SSL objectives. In this way, these two approaches may be complimentary.

**Data Augmentation in Feature Space**  There exist multiple works which can similarly be interpreted as performing data augmentation in feature space both for supervised and self-supervised learning. These include Dropout (Srivastava et al., 2014), Manifold Mixup (Verma et al., 2018), and others which perform augmentation directly in feature space (DeVries & Taylor, 2017; Hendrycks et al., 2020), or through generative models (Sandfort et al., 2019). We see that our work is fundamentally different from these in that it is not limited to simply performing an augmentation which would have been performed in the input in latent space. Instead, it maximally leverages structured representations to generalize all of these approaches and show how others can be included under this umbrella. Specifically, a framework such as DIM(L) is not explicitly performing an augmentation in latent space, but rather comparing two subsets of a representation which are offset by an augmentation. As we discuss in Section 6, this distinction is valuable for developing novel SSL algorithms which can substitute learned homomorphisms for learned augmentations – potentially sidestepping challenges associated with working in input-space directly.

**Hybrid A-SSL + F-SSL**  Some recent work can be seen to leverage both augmentation-based and feature-space losses simultaneously. Specifically, Augmented Multiview Deep InfoMax (Bachman et al., 2019) achieves exactly this goal and is demonstrated to yield improved performance over its non-hybrid counterparts. Although similar in motivation, and perhaps performance, to our proposed framework, the Homomorphic SSL framework differs by unifying the two losses into a single objective, rather than a sum of two separate objectives.

## 6    Discussion

In this work we have studied the impact of combining augmentation-homomorphic feature extractors with augmentation-based SSL objectives. In doing so, we have introduced a new framework we call Homomorphic-SSL which illustrates an equivalence between previously distinct SSL methods when the homomorphism constraint is satisfied.

Primarily, the H-SSL framework opens the door to further development of augmentation free self-supervised learning, as was initially pursued in the literature by frameworks such as Deep InfoMax (Hjelm et al., 2019). The necessity for hand-designed augmentations is known to be one of the current limiting factors with respect to self-supervised learning, contributing to biased representations (Xiao et al., 2021; Tsai et al., 2022; Dangovski et al., 2022) and more generally determining the overall generalization properties of learned representations (Ji et al., 2021; Wang et al., 2022; von Kügelgen et al., 2021). We therefore believe our work provides a valuable new alternative viewpoint through which these limitations may be addressed.

Secondly, with increased augmentation-free SSL methods arises the potential for layerwise 'greedy' SSL, as demonstrated by the Greedy InfoMax work (Löwe et al., 2019). We believe a factor which has to-date limited the development of layerwise self-supervised learning approaches has indeed been the dominance of augmentations in SoTA SSL, and a lack of knowledge about how these augmentations may be performed equivalently in the deeper layers of a neural network. Homomorphic feature extractors exactly provide this knowledge and therefore promise to extend the possibilities for such 'greedy' self-supervised learning methods which have significant implications for computational efficiency and biological plausibility.

**Future Work**  We believe that one of the most promising implications of the H-SSL framework is that the long-sought goal of 'learned SSL augmentations' may, in this view, be equivalently achieved through learned homomorphisms. While the field of learned homomorphisms is very new and undeveloped, we believe there are already interesting approaches in this direction which would provide an immediate starting point for future work (e.g. Keller & Welling (2021); Keurti et al. (2022); Connor et al. (2021); Dehmamy et al. (2021); Pal & Savvides (2018)). Since these approaches differ significantly from their input-space counterparts, it is likely that they may have the potential to circumvent otherwise difficult obstacles of operating in input-space.

**Limitations**  Despite the unification of existing methods, and benefits from generalization, we note that this approach is still limited. Specifically, the equivalence between $\mathcal{L}^{\text{A-SSL}}$ and $\mathcal{L}^{\text{H-SSL}}$, and the benefits afforded by this equivalence, can only be realized if it is possible to analytically construct a neural network which is equivariant with respect to the transformations of interest. Since it is not currently known how to

construct neural networks which are analytically equivariant with respect to all input augmentations used in modern SSL, this constraint is precisely the greatest current limitation of this framework. Although the field of equivariant deep learning has made significant progress in recent years, state of the art techniques are still restricted to E($n$) and continuous compact and connected Lie Groups (Finzi et al., 2020; 2021; Cesa et al., 2022; Weiler & Cesa, 2019). We believe in this regard, our analysis sheds some light on the success of methods which perform data augmentation over those which operate directly in feature space in recent literature – it is simply too challenging with current methods to construct models with structured representations for the diversity of transformations needed to induce a sufficient set of invariances for linear separability of classes. We therefore propose this work not as an immediate improvement to the state of the art, but rather as a new perspective on SSL which provides a bridge to previously distant literature.

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

## A   Experiment Details

**Model Architectures**   On MNIST (LeCun & Cortes, 2010), we used a backbone network $f$ composed of three SESN convolutional layers with # channels (32, 64, 128), kernel sizes (11, 7, 7), effective sizes (11, 3, 3), strides (1, 2, 2), padding (5, 3, 3), no biases, basis type 'A', BatchNorm layers after each convolution, and ReLU activations after each BatchNorm. The output of this final ReLU was then considered our $z$ for contrastive learning (with $\mathcal{L}_{A-SSL}$ and $\mathcal{L}_{H-SSL}$) and was of shape $(128, S \times R, 8, 8)$ where $S$ was the number of scales for the experiment (either 1 or 6), and $R$ was the number of rotation angles (either 1 or 4). For experiments where the transformation studied was not translation, we average pool over the spatial dimensions before applying the projection head $h$ to achieve a consistent dimensionality of 128. For classification, an additional SESN convolutional layer was placed on top with kernel size 7, effective size 3, stride 2, and no padding, thereby reducing the spatial dimensions to 1, and the total dimensionality of the input to the final linear classifier to 128.

On CIFAR10 we used a ResNet20 model composed of an initial SESN lifting layer with kernel size 7, effective size 7, stride 1, padding 3, no bias, basis type 'A', and 9 output channels. This lifted representation

was then processed by a following SESN convolutional layer of kernel size 7, effective size 3, stride 1, padding 3, no bias, basis type 'A', and 64 output channels. This initial layer was followed by a BatchNorm and ReLU before being processed by three ResNet blocks of output sizes (128, 256, 512) and initial strides of (1, 2, 2). Each ResNet block is composed of 3 SESN Basic blocks as defined here (`https://github.com/ISosnovik/sesn/blob/master/models/stl_ses.py#L19`). The output of the third ResNet block was taken as our $z$ for contrastive learning (again for $\mathcal{L}_{A-SSL}$ and $\mathcal{L}_{H-SSL}$) of shape $(512, S \times R, 7, 7)$. Again, as for MNIST, for experiments where the transformation studied was not translation, we average pool over the spatial dimensions before applying the projection head $h$ to achieve a consistent dimensionality of 512. For classification, the vector $z$ was first max-pooled along the scale/rotation group-axis $(S \times R)$, followed by a BatchNorm, a ReLU, and average pooling over the remaining $7 \times 7$ spatial dimensions. Finally, we apply BatchNorm to this 512-dimensional vector before applying the non-linear projection head $h$.

On Tiny ImageNet we use a Resnet20 model which has virtually the same structure as the CIFAR10 model, but instead uses 4 ResNet blocks of output sizes (64, 128, 256, 512) and strides (1, 2, 2, 2). Furthermore, each ResNet block is composed of only 2 BasicBlocks for TIN instead of 3 for CIFAR10. Overall this results in a $z$ of shape $(512, S \times R, 4, 4)$, and a final vector for classification of size 512. We note that we do not include Translation results in Table 1 for Tiny ImageNet precisely because the spatial dimensions of the feature map with this architecture are too small to allow for effective H-SSL training in the settings we used for other methods.

All models used a detached linear classifier for computing the reported downstream classification accuracies, while the Supervised baselines used an attached linear layer (implying gradients with respect to the classification loss back-propagated though the whole network). All models additionally used an attached non-linear projection head $h$ constructed as an MLP with three linear layers. For MNIST these layers have of output sizes (128, 128, 128), while for CIFAR10 and TIN they have sizes (512, 2048, 512). There is a BatchNorm after each layer, and ReLU activations between the middle layers (not at the last layer).

**Empirical Validation**  For the experiments in Table 1, we use two different methods for data augmentation, and similarly two different methods for selecting the representations ultimately fed to the contrastive loss for the A-SSL and H-SSL settings.

For A-SSL we augment the input at the pixel level by: randomly translating the image by up to $\pm$ 20% of its height/width (for translation), randomly rotating the image by one of $(0^o, 90^o, 180^o, 270^o)$ (for rotation), or randomly downscaling the image between 0.57 and 1.0 of its original scale. For S-SSL we use no input augmentations.

For both methods we use only a single fiber, meaning the base size $|\boldsymbol{g}_0|$ is 1. For A-SSL, we randomly select the location $\boldsymbol{g}_0$ for each example, but we use the same $\boldsymbol{g}_0$ between both branches. For example, in translation, we compare the feature vectors for two translated images *at the same pixel location*. Similarly, for scale and rotation, we pick a single scale or rotation element to compare for both branches. For H-SSL, we randomly select the location $\boldsymbol{g}$ independently for each example *and independently for each branch*, effectively mimicking the latent operator.

**H-SSL Without Structure**  In Table 2, we use the same overall model architectures defined above (3-layer model or ResNet20), but replace the individual layers with non-equivariant counterparts. Specifically, for the MLP, we replace the convolutional layers with fully connected layers with outputs (784, 1024, 2048). For the convolutional models (denoted CNN $(6 \times CHW)$), we replace the SESN kernels with fully-parameterized, non-equivariant counterparts, otherwise keeping the output dimensionality the same (resulting in the $6 \times$ larger output dimension).

Furthermore, for these un-structured representations, in the H-SSL setting, we 'emulate' a group dimension to sample 'fibers' from. Specifically, for the MLP we simply reshape the 2048 dimensional output to (16,128), and select one of the 16 rows at each iterations. For the CNN, we similarly use the 6 times larger feature space to sample $\frac{1}{6}^{th}$ of the elements as if they were scale-equivariant.

**Parameters of H-SSL**  For Figure 3 (left), we select patches of sizes from 1 to $|G| - 1$ with no padding. In each setting, we similarly increase the dimensionality of the input layer for the non-linear projection head $h$ to match the multiplicative increase in the dimension of the input representation $z(g)$. For the topographic distance experiments (right), we keep a fixed base size of $|g_0| = 1$ and instead vary the maximum allowed distance between randomly sampled pairs $g_1$ & $g_2$.

## B  Extended Background

**Related Work**  Our work is undoubtedly built upon the the large literature base from the fields equivariant deep learning and self-supervised learning as outlined in Sections 1 and 2. Beyond this background, our work is highly related in motivation to a number of studies specifically related to equivariance in self-supervised learning. Most prior work, however, has focused on the undesired invariances learned by A-SSL methods (Xiao et al., 2021; Tsai et al., 2022) and on developing methods by which to avoid this through learned approximate equivariance (Dangovski et al., 2022; Wang et al., 2021). Our work is, to the best of our knowledge, the first to suggest and validate that the primary reason for the success of feature-space SSL objectives such as DIM(L) (Hjelm et al., 2019) and GIM (Löwe et al., 2019) is due to their exploitation of equivariant backbones.

**DIM(L) in H-SSL**  In this section we outline precisely how the Deep Infomax Local loss DIM(L) relates to the H-SSL framework proposed in Section 3. Specifically, in Deep InfoMax (DIM(L)) the same general form of the loss function is applied (often called InfoNCE), but the cosine similarity is replaced with a log-bilinear model: $\text{sim}(a, b) = \exp\left(a^T W b\right)$. Additionally, and most importantly to this work, rather than computing the similarity between two differently augmented versions on an image, the loss is applied between different spatial locations of the representation for a single image, again with a head $h$ applied afterwards. If we let $g \sim \mathbb{Z}_{HW}^2$ refer to sampling a contiguous patch from the spatial coordinates of a convolutional feature map, we can write this general Feature-Space InfoMax loss ($\mathcal{L}_{\text{FSIM}}$) as:

$$\mathcal{L}_{\text{FSIM}}(\mathbf{X}) = -\frac{1}{N} \sum_i^N \mathbb{E}_{g_1, g_2 \sim \mathbb{Z}_{HW}^2} \log \frac{\exp\left(\text{sim}\left(h(z_i(g_1)), h(z_i(g_2))\right)/\tau\right)}{\sum_{k \neq i}^N \sum_{j,l}^2 \exp\left(\text{sim}\left(h(z_i(g_j)), h(z_k(g_l))\right)/\tau\right)} . \tag{8}$$

To show that this is equivalent to our $\mathcal{L}_{\text{H-SSL}}$, we see that the randomly sampled spatial patches $g_1, g_2$ can equivalently be described as a single base patch $g_0$ shifted by randomly sampled translations $g_1$ and $g_2$. Explicitly,

$$\mathcal{L}_{\text{FSIM}}(\mathbf{X}) = -\frac{1}{N} \sum_i^N \mathbb{E}_{g_1, g_2 \sim G} \log \frac{\exp\left(\text{sim}\left(h(z_i(g_1^{-1} \cdot g_0)), h(z_i(g_2^{-1} \cdot g_0))\right)/\tau\right)}{\sum_{k \neq i}^N \sum_{j,l}^2 \exp\left(\text{sim}\left(h(z_i(g_j^{-1} \cdot g_0)), h(z_k(g_l^{-1} \cdot g_0))\right)/\tau\right)} . \tag{9}$$

Thus, we see that Feature-Space InfoMax losses are included in our framework, and can therefore be seen to be equivalent to input-augmentation based losses with an equivariant backbone, where the set of augmentations is limited to the translation group $G \equiv \mathbb{Z}_{HW}^2$, and the $g_0$ base size is a single spatial coordinate ($|g_0| = 1$) rather than the size of the full representation ($|g_0| = |G|$).

## C  Broader Impact

This work is primarily related to understanding and improving self-supervised learning – a training method for deep neural networks which is able to leverage large amounts of unlabeled data from the internet, making it one of the most used methods for state of the art image and text generative models today (Radford et al., 2021; Ramesh et al., 2021). Such models have significant broader impact and potential negative consequences which are beyond the scope of this work. We refer readers to discussions of those paper for further information. Specifically, this work aims to improve such SSL techniques, thereby inheriting the broader impact of these models.

