# OpenReview forum: "Homomorphic Self-Supervised Learning"
_TMLR — Accepted by TMLR_

### Review · Reviewer_CJwi · 2023-06-21

**Summary Of Contributions:**

The authors introduce a formalism of Homomorphic SSL (H-SSL) and they show that this formalism can explain both Augmentation-based SSL (A-SSL) and Feature-space SSL (F-SSL). The SSL loss considered in the paper is contrastive-based (InfoNCE). Practically, in H-SSL we act on the output representations space to form positive pairs. Importantly, in order for this SSL to work we need a structural bias, i.e. the architecture we use should be equivariant. The authors support their claims with clear derivation and reasonable experiments. Another interesting take from the paper is a potential explanation of the success of SSL objectives based on CNNs, i.e. that the objectives might have been particularly suitable to the translation-equivariant bias of a CNN.

**Audience:**

Yes

**Claims And Evidence:**

Yes

**Requested Changes:**

CA. More discussion about the equivariant biases and the success of SSL. I feel like currently the discussion is hidden in the text, however, I think this is quite interesting.

CB. How would you explain the success of ViTs in SSL in the context of CA? I am somewhat reminded of that paper: https://openreview.net/forum?id=JL7Va5Vy15J. Curious to hear your take.

CC: I think an experiment with non-contrastive losses would be useful to see the empirical scalability of your approach.



**Strengths And Weaknesses:**

Strengths:

SA: H-SSL makes sense and it is a nice addition to the literature.

SB: The authors are mostly clear about the limitations of their approach.

SC: It is nice that H-SSL does not need augmentations on the input space.

Weaknesses:

WA. The authors use A-SSL on an equivariant neural network. Typically, in equivariant NNs, there is a global (group) pooling layer at the end, which makes the representation *invariant* to the group action. It seems that if one uses the pooling layer, then the objective of the positive pairs in contrastive learning objective is less clear. I don't think the authors discuss that detail, but it would be nice to clarify my confusion.

WB: What is the qualitative conclusion of Figure 3? Typically, Tian et al. found that there is a "Sweet spot" for augmentation strength. Is there such a "Sweet spot" in H-SSL?

WC: While H-SSL is input augmentation-free, the complexity is transferred to the architecture and the action on the output representations. This is addressed by the authors, but I would like to further underline it here.

WD: The GPU hours in the Training Details are another limitation that is buried in the text. It seems like training equivariant NNs at scale is very challenging, which makes me skeptical about the scalability of H-SSL to large (more realistic) SSL settings.

Minor:

MA. I would suggest moving Table 2 to the top. Currently, there is a paragraph between Tables 1 and 2, which breaks the flow of the text.

MB: In Figure 3 (right), the CIFAR curves seem to be at a different scale than the MNIST curves. It would be useful to plot a separate figure for CIFAR. For example the "Scale" (not to be confused with the *scale* [of the test set drop] in my comment) curve for CIFAR is flat currently.

MC: In Appendix A, when you cite Li et al. (2021) use \citep instead of \citet.

---

> ### Author Response · Authors · 2023-08-16
> **Response to Reviewer CJwi (1/2)**
>
> We thank Reviewer CJwi for their complete and thorough review of our paper, including their acknowledgement of the value of our contributions. Specifically, we are very grateful the reviewer has expressed interest in how our work may provide a partial explanation for the success of early SSL methods. We agree that this is one of the most exciting parts of our work, and that by formalizing it in this way, we hope to have facilitated future work in this direction.
>
> Below, we answer the reviewer’s questions and elaborate some points from our work.
>
> Response to WA: The reviewer is concerned that the global group pooling operation which is typically employed in equivariant neural networks would yield an invariant representation, making contrastive learning invalid. We agree that the reviewer’s intuition is right in this regard, however, let us clarify our experimental setup to resolve this concern. Specifically, to make the comparison between A-SSL and H-SSL fair, for both methods we sub-select a single group element (fiber) from the representation to use in the contrastive loss. For A-SSL these fibers come from two differently augmented images, for H-SSL the fibers are related by latent representation of the transformation $\Gamma_g$. Note that this is all done before any group pooling is applied. Only during evaluation (when we perform classification with a detached classifier) do we perform global group pooling to achieve invariance and improve classification accuracy. We have outlined this procedure in more detail in Section 4.1 and Appendix A, however we thank the reviewer for highlighting that our original text may have been unclear about this point. We will aim to clarify the text for the final version to make this more apparent.
>
> Response to WB: The reviewer questions the desired qualitative conclusions of Figure 3, specifically in relation to the results of prior work studying augmentation strength (Tian et al. 2020a). While we agree that our results in Figure 3 (right) do not appear to show a ‘sweet spot’ of the H-SSL parameters, as Tian et al. show for augmentation strength in A-SSL, we include these plots to demonstrate that the new parameters illuminated by the H-SSL framework do indeed have a significant impact of model performance and therefore may serve as valuable knobs for tuning (similar to how Tian et al. show that augmentation strength is important to performance). The reason that we mention the work of Tian et al. (2020a) specifically is due to the results from Figure 3 of that paper. This figure shows that patches of varying distances have differing amounts of mutual information as a function of distance, and this directly impacts downstream model performance. Since our results in Figure 3 with translation can be seen as experimentally identical (we compare representation patches at different translation distances), we found the comparison apt. The reason that we do not see such a nice inverted U-shaped curve as Tian et al. is that their models have been trained on the much higher resolution DIV2K dataset, allowing for patch offsets of up to 384 pixels. In our case, since we are working on the latent representations of low resolution MNIST and CIFAR10 images, we only have a maximum offset of 8-pixels, and therefore believe that we are only capturing a fraction of the curve illustrated by others. We thank the reviewer for bringing this up and we will add this discussion to the full paper.
>
> Response to WC: The reviewer is correct that the complexity of augmentations are effectively transferred to the encoder. We will underline this in the text. We would also like to note however, that this complexity is of a different sort (as elaborated in our response to Reviewer 7QDP), and therefore may open the door to novel approaches previously impossible with input augmentations.
>
> Response to WD: The reviewer raises concern that training of equivariant models is computationally expensive. While they certainly can be slightly more expensive due to expanded feature space dimensionality, this cost is typically not prohibitive for training on full scale datasets such as ImageNet. In practice, we found that equivariant models did not train more than a factor of 2-3 slower than their non-equivariant counterparts, dependent on architecture, and often nearly equivalently as fast as non-equivariant baselines. Conversely, however, H-SSL may provide computational complexity reductions in certain settings, since there is not a need to perform two forward passes on augmented images, and rather all contrastive learning can be performed in the feature space of a single image. With this new framework, sufficient future work on layerwise contrastive losses, or new contrastive losses, may uncover computationally cheaper SSL algorithms leveraging H-SSL. We will bring a discussion of this issue into the main text for clarity.

---

> > ### Author Response · Authors · 2023-08-16
> > **Response to Reviewer CJwi (2/2)**
> >
> > With respect to the changes requested by the reviewer, we provide the following additional discussion:
> >
> > Response to CA: As mentioned in our first paragraph of this response, we agree that our H-SSL framework does shed some light on the success of early self-supervised methods, and we find this to be both interesting and valuable. We will certainly expand on the fact that equivariant architectures were crucial to these methods whether this was known at the time of their development or not.
> >
> > Response to CB: We thank the reviewer for bringing up this interesting connection. Indeed we believe that there is likely a deeper connection between equivariance and the performance of SSL models, similar to the correlations of equivariance error and Imagenet test accuracy depicted in Figure 5 of Gruver et al., 2023. In the context of the H-SSL framework, we are not sure at this point how much more our work can contribute to understanding the success of ViT’s for SSL. Specifically, we are unaware of any ViT-SSL models which leverage latent representations of augmentations to perform SSL in feature space as we have proposed here. To the best of our knowledge, most ViT models are used as feature extractors, with SSL objectives operating on the outputs between two differently augmented images (i.e. A-SSL). In contrast, methods such as Deep Infomax operate on a single unaugmented image and instead leverage the equivariance of the feature extractor in order to formulate SSL objectives in the feature space of a single image directly (F/H-SSL). However, we could be highly interested if formal statements can be made about the benefits of equivariance when combined with input augmentation for SSL. Perhaps this is related to aliasing and equivariance as described in Gruver et al., and may serve to complement the analysis presented in our H-SSL work. We will add a discussion on this point to our final text.
> >
> > Response to CC: We appreciate the reviewer pointing out this additional experiment which could benefit our work. While we agree that non-contrastive losses like BYOL do work well with smaller batch sizes, and therefore may scale more favorably to state-of-the-art, we note that due to the current limitations on equivariant architectures (as mentioned in our work and again brought up by reviewer 7QPD) it is unlikely that H-SSL is currently able to scale to state-of-the-art performance using any SSL objective given. Therefore, we mainly intend this work as an introduction of a new framework for SSL with theoretical justification, and an empirical validation of the soundness of the theory rather than validation of scalability. However, we are happy to discuss this point further and perform additional experiments should the reviewer find them necessary for validating the claims of our paper.
> >
> > Finally, we appreciate the minor comments noted by the reviewer and we will certainly make these changes in our final draft.

---

### Review · Reviewer_7QDP · 2023-06-28

**Summary Of Contributions:**

This work studies self-supervised learning (SSL) from the perspective of homomorphic functions. The paper is motivated by the intuition that image transformations with group structure are an essential part of modern contrastive self-supervised learning, which leads to an investigation of group properties of widely used self-supervised learning methods. Given a network which satisfies a certain group equivariance (meaning that a group action on the input space has a compatible group action in the output space), one can define a Homomorphic SSL (H-SSL) objective which uses a contrastive loss of group actions in the output space. This is shown to be equivalent to augmentation-based contrastive learning (such as SimCLR) which uses the corresponding input-space group action as a way to generate augmentations. Furthermore, it is shown H-SSL encompasses patch-based feature space SSL methods such as CPC and Deep Infomax. Experimental results corroborate the equivalence of augmentation SSL and H-SSL by showing very similar results for CIFAR-10, MNIST, and TinyImageNet for downstream linear classification. Ablation studies using a network without equivariance shows that the equivalence breaks down and H-SSL has very poor performance when the input-space group action is not preserved by the network forward pass.

**Audience:**

Yes

**Broader Impact Concerns:**

Broader impacts are appropriately discussed.

**Claims And Evidence:**

Yes

**Requested Changes:**

I have the impression that more experiments could be conducted to demonstrate the equivalence of H-SSL and A-SSL. For example, given a network with identical weights, one could measure the correlation between hidden layer activations from group actions on the input space and output space respectively. There should be nearly perfectly correlated (up to numerical inaccuracies from aliasing/boundary etc. that upset the theoretical equivalence).

**Strengths And Weaknesses:**

**Strengths**
* The motivation and exposition in the paper are clear and understandable. Investigating the relations between SSL augmentation and network group structure is a relevant and interesting direction.
* The work provides a novel perspective for understanding feature-based SSL such as Deep Infomax and augmentation-based SSL such as SimCLR within the same framework. The key insight is that group structure in the input space can be studied by group structure in hidden layers or the output space as long as the network is equivariant to the group action.
* Results in Table 1 give interesting preliminary evidence of the equivalence between augmented SSL and H-SSL in practical settings, and the results in Table 2 clearly show that group equivariant network architectures are vital for effective performance of H-SSL.

**Weaknesses**
* The primary weakness is the limited scope of the method (which is acknowledged by the authors in the limitations section). The method requires special equivariant network architectures, which are currently limited to a small number of group actions. Therefore the method can currently only be applied to rotation, translation, and scaling augmentations. Strong SSL baselines like SimCLR typically rely on more aggressive augmentations like cutout and colorization which might not be easy to describe in terms of group action, and which do not have special equivariant architectures. Therefore, the method is unlikely to contribute to SOTA SSL results without further development of equivariant architectures to support the method.

---

> ### Author Response · Authors · 2023-08-16
> **Response to Reviewer 7QDP**
>
> We thank Reviewer 7QDP for taking the time to review our paper. Their summary is precise and demonstrates a clear understanding of the field and our submission. We appreciate their acknowledgement of the relevance and novelty of our work, specifically with respect to unifying an understanding of previously distinct models such as Deep Infomax and SimCLR.
>
> With respect to the scope of the method, we agree with the reviewer that by framing self-supervised learning in terms of equivariant feature extractors, our work is primarily limited by the state of the relatively young field of equivariant neural network architectures, and therefore cannot compete with SoTA currently. Importantly, however, we would like to highlight that it is precisely this alternative perspective on self-supervised learning that equivalently illuminates new perspectives on how the concept of ‘augmentations’ may be generalized, and therefore opens the door for significant future work. Specifically:
> 1. We believe that by allowing for self-supervised learning objectives to operate entirely within the hidden layers of a network, the continued development of computationally efficient and biologically motivated layerwise self-supervised objectives such as Greedy InfoMax (Lowe et al, 2019) now has stronger mathematical grounding.
> 2. One long sought goal of self-supervised learning is that of *learned augmentations*, potentially alleviating the need for hand-designed augmentation strategies which are known to bias the resulting features of SSL algorithms (Xiao et al., 2021; Tsai et al., 2022; Dangovski et al., 2022). However to date it has proven challenging to learn such augmentations in input space (see Blass et al., 2021; Shi et al., 2022). As the reviewer has pointed out, the key insight of our submission is that augmentation is equivalent to latent group representations when given a homomorphic feature extractor. This suggests that the goal of *learning augmentations* may be equivalently approached from the direction of *learning homomorphisms* (e.g. Keller & Welling, 2021; Keurti et al., 2022), as outlined in our future work section. This alternative approach may prove more fruitful, avoiding the pitfalls of input-space augmentations.
>
> Therefore, although equivariance is a limitation of our proposed approach, we believe it can also be seen as an alternative viewpoint offering its own strengths which have significant potential to be explored.
>
> Finally, we would like to thank the reviewer for the suggestion of additional experiments to validate the equivalence between H-SSL and A-SSL. Specifically, the reviewer has suggested we compare the impact of input and output group actions in terms of the resulting neural representations. Indeed, as the reviewer has suggested, there is likely to be nearly perfect correlation. In the equivariance literature, such experiments are often referred to as measuring the ‘equivariance error’ of a model and are common when proposing new equivariant architectures. For example, the network architecture used in our experiments is the Scale Equivariant Steerable Network (SESN, Sosnovik et al. 2020), and in Figure 2 of the original paper they measure this equivariance error as a function of different model hyperparameters, showing it to be consistently low. In our submission, since we were simply using this model directly (and indeed from their open-source code), we did not find it crucial to re-evaluate the equivariance error. If the reviewer would like us to include our reproduction of these results, or a reference to the prior work, we would be more than happy to do so.

---

### Review · Reviewer_yinW · 2023-08-07

**Summary Of Contributions:**

One way of looking at self-supervised learning is that it 'bootstraps' the process of learning informative representations by applying transformations to the input that leave some underlying task-relevant information unchanged. Although learning objectives in SSL are only approximately related to some downstream task, the representations learnt have proven to be powerful across different application areas and domains. Another line of work on group equivariant networks seeks to hard-code equivariance to transformations on the input -- the idea being that it is a more principled and complete way of doing data augmentation when the set of transformations form a group.

The main idea of the paper is that some ideas in self-supervised learning can be unified and generalized when seen through the lens of equivariant learning. More specifically, it studies SSL when the backbone feature extractor is a group equivariant network. The paper shows that a family of SSL loss functions can be unified and generalized with this viewpoint. First, in section 2.1 is covers the basic ideas behind GCNNs, and in section 2.2 it covers the basic ideas behind SSL. Two common losses in SSL are divided into two categories, one that relies on a vector similarity between pairs of examples and then has a siamese/drLIM type contrastive loss function (e.g. the paper quotes something similar to the neighborhood component analysis objective), and another which could be considered 'non-contrastive" losses. They could be considered non-constrastive since they don't use negative example as in contrastive setups and substitute it by some approprite regularization instead. Based on the relationship between the example pairs, the authors further divide methods into A-SSL (augmented SSL) and a slightly more general version i.e. F-SSL. Then a simple formulation inspired by GCNNs is considered next and named H-SSL. It is argued that H-SSL includes both F-SSL and A-SSL.

Experiments consider a backbone equivariant network. An example network considered, for instance, is both equivariant to 90 degree rotations and to scale. The selection of the augmentation also follows a straightforward (which is motivated using the language of fibers from the steerable CNNs paper). Finally a non-equivariant backbone H-SSL is also considered. Some results are reported on MNIST, CIFAR10 and tine ImageNet to validate the proposed method.

**Audience:**

Yes

**Claims And Evidence:**

Yes

**Requested Changes:**

I would love to hear from authors on what they think of additional experiments and could continue the discussion based on that.

**Strengths And Weaknesses:**

Strengths:
- The main idea of the paper is simple and well motivated: combine homomorphic feature extractors with augmentation-based SSL objectives.
- The intuition from their formulation, regarding the influence of base size and the group under consideration, is simple but interesting.
- The paper shows that two common formulations in the SSL literature can be seen using the lens of the proposed H-SSL framework.
- The authors discuss the limitations of their approach with candor.

Weaknesses:
- The main idea is actually quite simple and intuitive (and this is a good thing), however, it is then incumbent on the authors to include experiments that could convince the practitioner of the usefulness of this approach. The experiments give the impression that they cover the bare minimum needed to support the discussion in the paper, but they are far from compelling. What happens with continuous groups are considered? What if we considered problems in equivariant learning specifically and then did the augmentation along transformations that the backbone network is not made equivariant to, but is nonetheless a symmetry that exists in the data. What are the failure cases of such a SSL approach when there is a symmetry mismatch between the symmetry of the backbone network + augmentations and that of the data. How do such representations generalize? Give the approach is very intuitive, I would like to see at least some results supporting some of these points. Currently, I am finding it a bit difficult to get a take home message from the paper.

---

> ### Author Response · Authors · 2023-08-16
> **Response to Reviewer yinW (1/2)**
>
> We thank Reviewer yinW for spending time with our submission and writing such a clear and thorough summary. We further appreciate that they have noted the ‘very intuitive’ nature of our contributions, and hope that this is reflective of the quality of our presentation.
>
> With respect to the reviewer’s listed weaknesses, it appears that they are primarily concerned with the ‘take home message’ from the work, and what the potential applications or ‘usefulness’ of the Homomorphic-SSL framework may be. In brief, we believe there are a few different valuable insights which are immediately afforded by considering self-supervised learning through the lens of homomorphic feature extractors. We have listed a few of these in our response to Reviewer 7QDP, and we expand on some of these below:
> - Primarily, the H-SSL framework opens the door to further development of augmentation free self-supervised learning, as was initially pursued in the literature by frameworks such as Deep InfoMax. The necessity for hand-designed augmentations is known to be one of the current limiting factors with respect to self-supervised learning, contributing to biased representations (Xiao et al., 2021; Tsai et al., 2022; Dangovski et al., 2022) and more generally determining the overall generalization properties of learned representations (von Kügelgen et al., 2021;  Ji et al., 2021; Wang et al., 2022). We therefore believe our work provides a valuable new alternative viewpoint through which these limitations may be addressed, and we believe our experiments have validated that viewpoint for future study.
> - With increased augmentation-free SSL methods arises the potential for layerwise ‘greedy’ SSL, as demonstrated by the Greedy InfoMax work (Lowe et al. 2019). We believe a factor which has to-date limited the development of layerwise self-supervised learning approaches has indeed been the dominance of augmentations in SoTA SSL, and a lack of knowledge about how these augmentations may be performed equivalently in the deeper layers of a neural network. Homomorphic feature extractors exactly provide this knowledge and therefore promise to extend the possibilities for such ‘greedy’ self-supervised learning methods which have significant implications for computational efficiency and biological plausibility.
> - As noted in our response to Reviewer 7QDP above, and briefly in our future work statement of the paper, we believe that one of the most promising implications of the H-SSL framework is that the goal of ‘learned SSL augmentations’ may, in this view, be equivalently achieved through learned homomorphisms. While the field of learned homomorphisms is very new and undeveloped, we believe there are already interesting approaches in this direction which would provide an immediate starting point for future work (e.g. Keller & Welling, 2021; Keurti et al., 2022). Since these approaches differ significantly from their input-space counterparts, it is likely that they may have the potential to circumvent otherwise difficult obstacles of operating in input-space.
>
> Overall, we therefore believe the main contribution and ‘take home message’ of this work to be the solid foundation we have introduced for future work on augmentation-free SSL. As outlined above, we believe this future work has significant potential in a variety of directions. While we appreciate that the reviewer considers the H-SSL framework itself is intuitive, we believe the future directions illuminated by the formalization of this framework to be less immediately obvious, thereby endowing our contributions with their value.
>
> Finally, with respect to this submission, while we agree that demonstration of the above potentialities arising from our framework would be highly interesting and impactful, we believe (as pointed out by the reviewer) that our current work does satisfy the minimal requirements for validating our main claims, and therefore provides a solid and valuable foundation for future work in this direction. With respect to this publication venue, we therefore would like to raise a discussion on whether the reviewer believes we have satisfied the main criteria for acceptance put forth by TMLR Guidelines, namely: “Are the claims made in the submission supported by accurate, convincing and clear evidence? This is the most important criteria. This implies assessing the technical soundness as well as the clarity of the narrative and arguments presented.”

---

> > ### Author Response · Authors · 2023-08-16
> > **Response to Reviewer yinW (2/2)**
> >
> > With respect to the reviewer’s specific questions, we provide preliminary answers below and welcome further discussion for any open questions.
> >
> > > What happens with continuous groups are considered?
> >
> > As long as there is a known representation of the group action in the latent space of the network (i.e. $\Gamma_g$), it is possible to use this representation to perform the equivalent ‘augmentations’ in latent space. This is equally true for networks which are equivariant with respect to discrete or continuous groups. Although this may change the notation used to describe H-SSL (e.g. Equation 7), it will have no impact on the resulting performance of the model due to the nature of equivariant neural networks commuting with group actions.
> >
> > > What if we considered problems in equivariant learning specifically and then did the augmentation along transformations that the backbone network is not made equivariant to, but is nonetheless a symmetry that exists in the data.
> >
> > In such settings, one would be able to use H-SSL for the equivariant augmentations and A-SSL methods for the non-equivariant augmentations, within the same network. This is indeed how extensions of the Deep InfoMax framework work built (e.g. Augmented Multiscale Deep InfoMax (AMDIM), Bachman et al. 2019), and were demonstrated to work well.
> >
> > > What are the failure cases of such a SSL approach when there is a symmetry mismatch between the symmetry of the backbone network + augmentations and that of the data.
> >
> > To be clear, if there is not a known representation of the group action in the feature space of the deep neural network (i.e. it is not equivariant w.r.t. the augmentation) it will not be possible to use H-SSL methods to leverage the augmentation (without otherwise ‘learning the homomorphism’). As described above, it is likely still possible to use A-SSL methods within the same network to leverage such augmentations, and it is unlikely that there will be significant unforeseen consequences of such a combination. For example, a large set of network architectures used for SoTA SSL to-date are translation equivariant, yet they leverage A-SSL with respect to other (non-equivariant) augmentations without issue.
> >
> > > How do such representations generalize?
> >
> > In our work we have performed all experiments on the test set in the standard self-supervised learning ‘transfer’ setting. In this way, we have attempted to demonstrate that H-SSL methods generalize in the exact same way as A-SSL methods. Theoretically, these representations should similarly not differ in any way (as outlined by our derivation), except through the difference in backbone feature extractors compared with standard SSL.
> >
> > We thank the reviewer for bringing up these points which we agree are valuable to discuss. We propose to highlight them in the discussion section of our final work, and invite further discussion here for any points which still lack clarity.

---

### Decision · Action_Editors · 2023-09-30

**Recommendation:** Accept as is

**Comment:**

This paper explores self-supervised learning (SSL) using homomorphic functions and group-equivariant networks. It introduces a Homomorphic SSL (H-SSL) framework, highlighting the importance of group structures in image transformations for contrastive SSL. The study establishes an equivalence between H-SSL and augmentation-based SSL through experiments. Reviewers acknowledge that the claims have been well-supported with contributions to a novel SSL perspective, and appreciate the clarity and motivation. Somehow they express concerns about the method's limited scope due to special equivariant architectures, suggesting more experiments to enrich the paper.

**Audience:**

Yes.

**Claims And Evidence:**

Yes.